# Effects of Sucrose Replacement by Polyols on the Dough-Biscuit Transition: Understanding by Model Systems

**DOI:** 10.3390/foods12030607

**Published:** 2023-02-01

**Authors:** Mathilde Roze, Guénaelle Diler, Bruno Pontoire, Bruno Novalès, Camille Jonchère, Doina Crucean, Alain Le-Bail, Patricia Le-Bail

**Affiliations:** 1ONIRIS, UMR GEPEA CNRS 6144, Rue de la Géraudière, CS 82225, 44322 Nantes, France; 2INRAe, UR BIA 1268, Rue de la Géraudière, CEDEX 03, 44316 Nantes, France; 3USC 1498 INRAE-TRANSFORM Department and GEPEA UMR CNRS 6144, Rue de la Géraudière, 44316 Nantes, France

**Keywords:** biscuit dough, sucrose, maltitol, sorbitol, starch gelatinization, model systems

## Abstract

This study investigated the impacts of the complete substitution of sucrose by maltitol and/or sorbitol on the dough-crumb transition in biscuits. To this end, the phenomena of starch gelatinization/melting were studied at different moisture contents, both in the biscuit dough and model systems, by X-ray diffraction (XRD), differential scanning calorimetry (DSC), and by environmental scanning electron microscopy (ESEM). Observation of doughs in ESEM revealed sorbitol had a structure very different from sucrose and maltitol crystals. After forming the dough pieces, it could be seen that at least some sugar and maltitol crystals were still present while sorbitol flakes were solubilized. At a limiting real water content (~20% dry basis), adding sweeteners to the mixture increased the gelatinization temperature, more markedly for sucrose and maltitol, as well as increasing the enthalpy. These results were confirmed by the model systems analyses. The calorimetric study with mixing batch cells revealed that sorbitol dissolved completely while maltitol and sucrose competed with the flour constituents to capture water. The proportion of water available for the sorption of the starch grain and its gelatinization was therefore different according to the affinity of the sweetener for water, and might influence the degree and temperature of starch gelatinization/melting.

## 1. Introduction

Considering the strategies for reducing sugars in the food offer, polyols figure prominently because of their properties, which are advantageously similar to those of sugars. Polyols have several benefits, with a view to reducing sugar in cereal products in particular. Functionally, polyols perform the roles of sugars, with respect to texture and structure. Some polyols have a sweetening power relatively close to that of sucrose, although most of them are lower. In addition, they cause little or no cariogenesis, their energy value is lower compared to sugars, and some have an interesting glycemic index. 

When it comes to replacing sugars in cereal products, the question of their interactions with starch arises. Indeed, water-starch-sugar interactions are of primary importance in this type of product, and have consequently been the subject of numerous studies in the past [1,2,3,4,5,6,7,8]. Most of this work focuses on the effects of sugars on starch gelatinization in model systems most often composed of starch, water and a sugar. The authors have shown that sugars shift the starch gelatinization temperature. Several hypotheses have been put forward to explain this effect. They relate to modifications of the a_w_ of the plasticizing solution, modifications in the interactions between the constituents of the plasticizing solution and the molecules present in the starch grain, and even a modification of the glass transition temperature of the amorphous regions of the grain [9]. Two recent studies rather tend towards the theory of plasticizer-starch chain interactions [8,9], however this debate is still ongoing and the origin of this phenomenon is not yet fully resolved. 

Despite the plethora of publications on this subject, studies conducted on real products are scarce and lack in recency. Abboud and Hoseney [10] investigated the gelatinization of starch in a simplified cookie dough (without fat or leavening agents) in the presence of sugars, and concluded that there was no starch gelatinization in cookie dough. Varriano-Marston et al. [11] studied starch gelatinization in different cereal products from a methodological perspective by comparing microscopy, viscosity, and enzymatic and crystallographic analyses. They highlighted a partial gelatinization of starch in cookies. Regarding methods, they concluded that the analysis of viscosity was not a reliable method, but a combination of other methods (crystallography, microscopy) was relevant for the study of starch gelatinization in grain products. More recently, few studies comparing the effects of polyol-type substitutes on gelatinization in real doughs have been published. Psimouli and Oreopoulou [12] showed in cakes a relatively similar behavior of polyols compared to sucrose, with respect to starch gelatinization with the exception of mannitol, for which gelatinization is substantial and takes place at a lower temperature. Similarly, Martinez-Cervera et al. [13] did not show any effect of the formulation on the starch gelatinization enthalpy, but showed a greater gelatinization temperature shift in the presence of disaccharide polyols compared to monosaccharide polyols. To our knowledge, the only study conducted on low-moisture products, such as biscuits, mostly concludes that the partial and/or total substitution of sucrose by erythritol and maltitol has no significant effect on the temperature or starch gelatinization enthalpy [14]. 

The present study aims to characterize more finely the effects of a complete substitution of sucrose by maltitol and sorbitol on the dough-to-biscuit transition. This investigation was conducted through the analysis of biscuit dough and through the analysis of a series of model systems composed of flour, water and sweeteners in different proportions at real water content, but also in excess water, to link with existing literature on starch, mostly conducted at high water content. This analysis was carried out combining different methods, such as DSC, XRD and SEM. The priorities between ingredients for water were also studied with mixing cells in order to highlight the mechanisms involved.

## 2. Materials and Methods

### 2.1. Materials

#### 2.1.1. Ingredients

The biscuit doughs were prepared with type 65 wheat flour (Moulins Evelia—Terrena, France) (14.7% moisture, 9.5% protein, 0.61% ash), butter (Président, Lactalis, France), salt, water, sodium bicarbonate (Brenntag, France) and disodium pyrophosphate (Budenheim, Germany). Either sucrose (Béghin Say, Tereos, France), maltitol (Louis François, France) or sorbitol (Louis François, France) were used as sweeteners. The three formulations are presented in Table 1. Sweetener substitution was total and made on weight basis.

#### 2.1.2. Model System

Water contents of the main ingredients used in doughs and model systems were determined in order to standardize the final water content of mixtures. 4 g of sample were dried in an oven (Memmert, Germany) at 105 °C during 24 h. Each measurement was made in triplicate. Mean water contents are presented in Table 2.

The model systems were made from flour (F), water (W) and sweeteners (sucrose (Su), maltitol (Ma), sorbitol (So) and maltitol-sorbitol mixture (MaSo)). Ingredients used in model systems were the same as in biscuit doughs. The proportions of the various constituents were calculated from their initial water contents, the proportions in the real dough on a dry basis and the desired final water content (Table 3). The nomenclature system is as follows. For a model system consisting of flour (F), sucrose (Su) and water (W), for example: F_x_ (Su_y_. W_a_) with:

F_x_ flour content wet basis;

Su_y_ sugar content wet basis;

W_a_ amount of water added.

Parentheses specify which ingredients were mixed first. In the example, water and sugar were first mixed before adding flour. The total amount of water (80%, 50% or 20%) therefore includes the water present in each of the ingredients and the added water. A series of model systems including butter were also carried out in order to evaluate its impact on starch gelatinization. Since no effect on starch gelatinization was observed, these results are not presented.

The method of preparation of the model systems represents the mixing conditions of the actual dough. 10 g were prepared in a closed vial for each model system. First, some ingredients were manually pre-mixed for 3 min at room temperature (according to the previously decided order of incorporation); then, the other ingredients were added and the whole was mixed again for 3 min. The model system was left to rest for 30 min, before being rehomogenized and sampled for calorimetric and X-ray diffraction analyses. 

#### 2.1.3. Biscuit Dough Preparation

The day before production, all ingredients were stored at 23 °C to ensure similar temperatures for each production, except for butter, which was put to defrost from −18 °C to +15 °C. Butter, leavening powders and the sweeteners were mixed with warm water (50 °C) in a VMI BV422 kneader (VMI, France) at 62 rpm for 1 min, and 125 rpm for 2 min. Flour was added in a second step and mixed with the rest at 62 rpm for 1 min, then at 125 rpm for 2 min. Dough was left to rest for 30 min at 30 °C and dough pieces were formed with a Padovani R2 rotary moulder (Padovani, Italy). The average mass of each biscuit was 12.0 ± 0.3 g.

### 2.2. Dough Structure as Observed with Environmental Scanning Electron Microscopy 

#### 2.2.1. Dough Water Content

Similarly to ingredients, the dough water content was determined using an oven (Memmert, Germany) at 105 °C during 24 h. Measurements were made in triplicate, using 4 g of sample. Means and standard deviations were calculated and are presented in Table 4.

#### 2.2.2. Observation of Sweeteners in SEM

Sweetener powder pictures were taken with the FEI-Thermo Fischer Quattro S microscope without prior preparation. The powders were deposited in a thin layer on sample holders covered with a double-sided carbon pellet and placed in the chamber of the apparatus. The imaging was done by the detection of secondary electrons in a degraded vacuum (controlled pressure of 130 Pa) and the applied acceleration voltage was 10 kV. Working distance was set at 7 mm.

#### 2.2.3. Observation of Biscuit Dough in ESEM

The same microscope was used in “environmental mode” for the morphological observations of the biscuit dough. In environmental mode, the microscope chamber was filled with water vapor, allowing hydrated samples to be observed at pressures ranging from 400 to 1500 Pa, depending on temperature. It was thus possible to observe the more hydrated (about 20%, dry basis (d.b.)) doughs and dough pieces without preparation (i.e., in their “natural” state), while limiting the evaporation of humidity present in the samples, which could modify their structure during the observation. Several observations were made on several blocks of dough and on several dough pieces in different areas. Observations were made at +4 °C in environmental mode over a pressure range from 610 to 730 Pa, corresponding to a relative humidity ranging from 75 to 90% in the chamber, and with an acceleration voltage of 10 kV.

### 2.3. Study of Major Ingredient Interactions in Dough and Model Systems 

#### 2.3.1. Wide-Angle X-ray Diffraction Analysis

X-ray diffraction (XRD) was used to study the impact of substituting sugar with sorbitol and maltitol on the structural evolution of the starch during baking. For each formulation, about 50 mg of dough were placed in a copper ring between two sheets of mica to prevent water evaporation during heating kinetics placed in a HF591 heating cell (Linkam Scientific Instruments, Redhill, UK). X-ray diffraction was performed at wide angles with a Bruker’s D8 Discover spectrometer (Bruker AXS, Karlsruhe, Germany), combined with a 2D detector Vantec 500. Measurements were carried out at 40 kV and 40 mA with a copper tube (Cu Kα radiation of wavelength λ = 1.54059 Å) parallelized with a double Göbel mirror parallel optics system and collimated to produce a 500 μm diameter beam. Data were monitored in the direct beam position. The heating kinetics applied to the sample was 1 °C/min from +10 °C to +120 °C. Every 5 °C, a 10-min plateau was introduced to allow acquisition of the diffraction spectrum. The obtained spectra were normalized between 3 and 28 ° in 2θ. The same analysis was carried out in excess of water on different doughs using a 1.5 mm diameter capillary, sealed with wax to prevent water evaporation. 

#### 2.3.2. Differential Scanning Calorimetry 

DSC experiments were carried out using a multi-cell microcalorimeter MC DSC (TA Instruments, New Castle, DE, USA), and the data were analyzed using TA Universal Analysis 2000 software (version 4.5). The three cells were filled with homogeneous samples (either dough or model system). An empty pan was used as a reference. Cells were hermetically sealed with heat-resistant seals and heated from +10 °C to +120 °C at a heating rate of 1 °C.min^−1^, and cooled from +120 °C to +20 °C at a cooling rate of 2 °C.min^−1^. Concerning the doughs, the analysis was carried out on three different samples. The same experiment was carried out in the presence of excess water with three additional doughs, specifically at 70% water content, instead of 80%, to improve endotherm features for the enthalpy analysis. Thermograms were recorded and analyzed to determine the peak temperature (T_peak_) and the enthalpy (ΔH in J/g dry flour) of starch gelatinization. Results for each formulation are expressed as means of the nine cells with standard deviations. Calorimetric analysis focused on starch gelatinization/melting phenomena in dough and model systems. Enthalpies were therefore computed, according to the water content, either on the gelatinization endotherm commonly identified as G (dough and model system with excess water), or the dual gelatinization endotherm G+M_1_ (model systems with intermediate water content), or the M_1_ endotherm alone, corresponding to starch melting in systems with limiting water content. The M_2_ peak corresponding to the phase transition in amylose-lipid complexes was not characterized, since it was only observed in moderate-to-high water content systems and with no impact of the formulation on it. 

#### 2.3.3. Mixing “Batch” Cells

Specific mixing cells were used to characterize the water uptake priorities between the flour and the different sweeteners (Figure 1). For this, a flour-sweetener mixture with a ratio of 1:1 was introduced into the bottom compartment (1) of the “batch mixing” cell (SETARAM, Caluire, France). Water was injected into the upper compartment (2) using a syringe. The sample and the reference cells were placed in a µDSC 7 evo microcalorimeter (SETARAM, Caluire, France) in isotherm mode and left at equilibrium at 20 °C for 30 min; then, the water was released into the lower compartment by a rotating rod mechanism.

### 2.4. Statistical Analysis

Each dough formulation and each model system were produced in triplicate. For each measurement, three samples per replicate were collected. Results are presented as mean value of the nine samples ± standard deviation. Statistical significances of the differences were computed with a one-way ANOVA combined to a Duncan test (α = 5%) using R software (version 3.6.2) and agricolae R package (version 1.3.3).

## 3. Results and Discussion

### 3.1. Surface Aspect and Structure of Doughs

#### 3.1.1. Sweetener Features

Microscopic observation of the different sweeteners revealed their relatively different structures, in particular for sorbitol. Sucrose exhibited very geometric crystals, with 8 to 10 faces of different particle sizes (Figure 2a). Maltitol also presented very geometric crystals of a quite different shape, but of the same order of magnitude (Figure 2b). Sorbitol had a very different structure with larger agglomerates compared to sucrose and maltitol, revealing a fibrous appearance at higher magnification, similar to very fine agglomerated flakes (Figure 2c). Thus, the contact surface is much larger for sorbitol than for maltitol and sucrose.

#### 3.1.2. Loose Dough

The loose dough was observed by SEM in Low-Vacuum mode. The images obtained are shown in Figure 3. Observation of the control dough clearly revealed the starch grains embedded in the matrix (Figure 3a). Sucrose crystals were present (Appendix A) but not easily identifiable, as they were covered/caught in the matrix. The geometric facets were therefore less obvious to locate. However, bringing the beam closer to the surface of the sample confirmed the nature of the crystals by melting them.

The maltitol dough presented the same general structure, again with few crystals of maltitol undissolved in the matrix (Figure 3b and Appendix A). The maltitol dough had a “more powdery” appearance compared to the control dough, which appeared more particulate.

In the case of the sorbitol dough, the matrix was similar to that of maltitol, i.e., slightly less smooth and cohesive than the sucrose dough. Several whole aggregates of sorbitol were visible on the surface of the dough, but more frequently, “layers” of sorbitol flakes were observed (Figure 3c), spread on the matrix or embedded in it. The distribution of these flakes was not homogeneous. This SEM observation of the dough indicates that after kneading, none of the three sweeteners was fully dissolved in the dough.

#### 3.1.3. Dough Pieces

A second observation was made on dough pieces, that is to say, after the loose dough rested for 30 min and passed through the rotative press. The dough pieces were observed in environmental mode with the same scanning electron microscope at +4 °C between 610 and 730 Pa. 

The observation of the control dough pieces showed a slightly aerated dough, despite the compression due to the passage through the rotary moulder. Sucrose crystals as well as filaments were observed in the dough (white arrows in Figure 4a,d). These filaments attest a very slight development of gluten, due to the forces exerted on the dough by the machine rolls. 

The maltitol doughs were very similar to the control ones, although slightly more compact. Again, the dough was “drier” on the surface and maltitol crystals were visible (white arrow in Figure 4b). In the case of sorbitol, the observation did not show any flakes of non-solubilized sorbitol on the various dough pieces. For the two formulations without added sugars, fewer filaments were observed on the surface of the dough pieces.

Between the kneading and forming steps, it seems that the sorbitol had completely solubilized, since it was not detected in the dough pieces, while maltitol and sucrose still appeared, at least partially, in the crystalline state before baking.

### 3.2. Impact of Sugar Substitution on Starch Gelatinization in Biscuit Dough

#### Starch Transition in Dough during Heating

XRD

Doughs were analyzed by X-ray diffraction at wide angles while heating kinetics (20-120 °C) were applied. Observations were made on doughs at actual water content, that is, around 17.1 ± 0.6% w.b. (Table 3), and on doughs in excess water (70%.) Analysis of the diffraction patterns of the biscuit doughs revealed the peaks expected in the presence of wheat starch (type A) at 2θ = 15°, 17°, 18° and 23° (Figure 5). 

These peaks began to decrease from 90 °C in the control dough at actual water content. The appearance of the diffractograms was very similar between the control recipe and the maltitol recipe, while the 50% Malt-50% Sorb and sorbitol formulations seemed more distant. The peaks sagged more and earlier in temperature (from 80 °C). The presence of still very marked peaks at 120 °C indicates that starch gelatinization was only partial at this water content, especially for maltitol and sucrose doughs. 

In excess water content, gelatinization began at 60 °C for all four doughs. Loss of crystallinity was complete at 65 °C, as all the peaks were completely smoothed on the different diffractograms until 95 °C. At this water content, no delay in gelatinization was observed for the maltitol and control doughs.

These results suggest that the substitution of sucrose by sorbitol could induce a different behavior of starch when heated during baking.

DSC

The differential enthalpy analysis made it possible to study more precisely starch gelatinization in biscuit dough in the presence of the various constituents, particularly sucrose or its substitutes. Again, this study was carried out on dough with an initial water content, but also on dough with excess water, in order to compare with results already available in the literature. 

The differential enthalpy analysis of doughs with excess water makes it possible to highlight and quantify the complete gelatinization of the starch. Starch gelatinization was materialized by an endothermic peak on the thermogram around 63 °C for the formulations without added sugars and the reference formulation (Figure 6). The peak temperature values are given in Table 5. For all formulations, sorption and swelling began at just over 56 °C and gelatinization was complete at around 77-78 °C. The peak temperature varied between 62.7 °C for the sorbitol dough and 63.8 °C for the control dough. These results are in agreement with our findings obtained with XRD analysis.

For a limiting water content (initial dough water content < 20%), gelatinization began later (Figure 7), irrespective of the formulation. The gelatinization peak appeared at around 96.6 °C for the sorbitol dough, while the substitution of sucrose by maltitol did not shift the gelatinization peak (T_peak_ ≈ 108.6 °C). These observations are in agreement with the results obtained for maltitol and control doughs by Laguna et al. [14]. Psimouli and Oreopoulou [12] also observed a smaller shift of starch gelatinization temperature with sorbitol compared to sucrose and maltitol in cake batters. The starch gelatinization peak in the 50%Malt-50%Sorb dough is between the two single-polyol doughs at 103.9 °C.

Sucrose substitution did not seem to have an effect on enthalpy, except for the sorbitol dough, which exhibited a slightly higher enthalpy (0.38 J/g dry flour (d.f.) vs. 0.30 J/g d.f.). According to Allan et al. [8], the magnitude of the T_peak_ shift could depend on its physicochemical characteristics (degree of polymerization, spatial arrangement, etc.). The effect is assumed to be correlated with the number of carbon atoms and, more generally, the molecular weight of solutes and their ability to interact with starch molecules and water. Previous studies proved that there is a smaller delaying effect on starch gelatinization for monosaccharides compared to disaccharides or trisaccharides [1,2,6,7,8]. As sorbitol is a monosaccharide, while maltitol and sucrose are disaccharides, our results would agree with this hypothesis explaining why starch melting occurred at lower temperatures.

### 3.3. Insights Extracted from Model Systems

Interactions of Ingredients and Impact on Starch Gelatinization

Water Content Effect

In order to characterize the interactions between the main ingredients in the dough and their impact on starch gelatinization, a series of model systems was made from flour, sucrose, maltitol and sorbitol at different levels of moisture. These model systems were studied by calorimetry. The values of enthalpies and temperature of onset, peak and end of gelatinization are presented in Table 6. Figure 8 shows the general appearance of the thermograms obtained for the flour-water systems at 80, 50 and 20% of water content (model systems no. 1, 2 and 3). 

In non-limiting moisture content, starch gelatinization was materialized with a peak at 60.3 °C. A second weaker peak around 95 °C was visible on the thermograms. This peak corresponds to the melting of the amylose-lipid complexes. The mean starch gelatinization enthalpy was 6 J/g d.f.

At intermediate water content, the thermogram showed the same peak at around 61.1 °C but exhibited an associated shoulder. The second peak was visible around 110 °C. Due to this slightly different profile, the integration of G + M_1_ peaks led to greater enthalpies than in the previous case. At this moisture content, it is therefore appropriate to compare the formulations with each other, and not to comment on the values in absolute terms.

Starch gelatinization is often described in two different events: (a) water sorption leading to hydration of amorphous parts in the starch granule and (b) destructuration of starch crystallites further associated with leaching of amylose chains from the starch granule. According to Eliasson A.C. [15], in excess water, these two steps are very quick and occur simultaneously, but when water content is limiting, the events are slower and usually divide into two distinct endotherms on DSC thermograms [10,16,17].

With very limiting water content, reading thermograms becomes much more complex. Indeed, at this water content, the starch gelatinization was very limited. The event should be defined as starch melting, as the endotherm M_1_ was barely visible on the thermogram. A single melting peak appeared at a higher temperature (79.9 °C). The peak corresponding to the fusion of the amylose-lipid complexes was not visible, as it was pushed out of the measurement range (>120 °C). The enthalpy testified to the very limited nature of the event in this case: barely 0.2 J/g d.f. or 3% of the enthalpy obtained in excess water.

Effects of Sweeteners

Figure 9a shows the general appearance of the thermograms obtained for the systems in excess water, namely, n°1, 4, 7, 10 and 13, corresponding to the mixtures with flour only, flour-sucrose, flour-maltitol, flour-50% malt–50% sorb, and flour-sorbitol.

The model systems with sweeteners have the same shape of thermogram as the system with flour alone. The gelatinization peak was located at around 60 °C. Sucrose and maltitol appear to have shifted the peak very slightly from 60.3 °C to 61.5 °C, however this shift was not significant. Regardless of the sweetener, there was no effect on peak width (T_end_ – T_onset_).

Regarding enthalpies, the addition of sweeteners appears to have increased the extent of gelatinization to varying degrees, even though differences were not significant. Similar results have been reported by Jang et al. [4] in starch-sucrose systems between 40% and 60% moisture contents. 

Figure 9b shows the general appearance of the thermograms obtained for the 50% water systems, i.e., mixtures no. 2, 5, 8, 11 and 14. A stronger peak shift was observed with the addition of sweeteners at this water content. The gelatinization peak was located at 67.9 °C in the case of sucrose and maltitol, and at 65.9 °C in the case of sorbitol. The latter seemed to offset the starch gelatinization temperature significantly less, a phenomenon already observed in biscuit dough. The thermograms of flour-sweetener-water mixtures had a slightly less extensive endotherm (~27 °C) than those of flour-water mixtures (~36 °C); the shoulder appeared to be less separated from the peak. This phenomenon has been observed by Eliasson [3] and Ghiasi, Hoseney and Varriano-Marston [18], who attributed it to a lower viscosity of the starch-water-sugar mixture compared with the starch-water mixture. Thus, the solvent volume is increased as water and dissolved sucrose are considered, and not only water volume. The addition of sweeteners did not shift the melting temperature of amylose-lipid complexes, however this peak was affected by the lower water content.

The impact of sugars on the starch gelatinization temperature has already been demonstrated on starches from several botanical sources and in the presence of different types of sugars [1,2,3,4,7]. At first glance, the effect of adding sugars is similar to the effect of lowering the water content: the starch gelatinization is shifted to higher temperatures. A similar effect has been demonstrated in the presence of polyols whose properties are close to sugars [6,8]. This is confirmed by our results, which show a shift in gelatinization temperature, both with the addition of sucrose and with the addition of maltitol and sorbitol. As mentioned previously, this effect is modulated by the molecular weight of the solutes, an element that seems to be confirmed in our tests, as sorbitol steadily exhibits a slightly less marked shift.

Different hypotheses have been put forward in previous work to explain the shift observed in the presence of sugars: effects on water activity [19,20], effects on the glass transition temperature of the plasticized amorphous region [21], and impacts on intermolecular hydrogen bonds [1,5,22,23]. The most recent study by Allan et al. [8] investigated the effects of different sugars on the gelatinization temperature of wheat starch and concluded that sweeteners increase the starch gelatinization temperature by forming intermolecular interactions with starch in the amorphous regions.

The gelatinization enthalpies were again greater in the case of three-component mixtures (systems no. 5, 8, 11 and 14) compared to the flour-water mixture (system no.2), but did not differ depending on the sweetener used. 

Figure 9c shows the general appearance of the thermograms obtained for the limiting water systems no. 3, 6, 9, 12 and 15. As with the observations of doughs at real water content, the analysis of the thermograms is more complex. The endotherms associated with starch melting were hardly visible. These were more shifted towards high temperatures and were of smaller size. The differences were then more marked. The peak shifted from 60 to 80 °C in the case of the flour-water system when decreasing water content from 80% to 20%, while the peaks shifted from 61 °C to 105 °C approximately for flour-sugar-water and flour-maltitol-water systems. Sorbitol had a smaller offset: from 60 °C to 93 °C.

The enthalpy values obtained attest to the very partial nature of the starch gelatinization. The highest enthalpy was observed for sorbitol (12-F_7.13_(So_1.84_ W_1.03_)), although it exhibited greater variability. This difference in enthalpy could possibly result from the concomitant fusion of sorbitol, or could be associated with the smaller shift in starch gelatinization temperature.

The thermograms of systems containing maltitol alone or as a mixture (systems no.9 and 15) also presented an endotherm at a lower temperature, around 50 °C (Figure 9d). This peak was not observed at a higher water content and might result from the solubilization of maltitol. It is interesting to note that, despite very strong similarities in the physicochemical properties of maltitol and sucrose, the thermograms of the sucrose mixtures did not exhibit this endotherm.

In the same way as for the dough with real water content, the peaks corresponding to the melting of the amylose-lipid complexes were no longer visible in the measuring range.

The study of the model systems confirmed the observations made on the dough, whether in excess water or at initial water content. Our results showed that the sweetening effect becomes more marked as the water content decreases, confirming a very different feature when sucrose is substituted by sorbitol, while the substitution by maltitol exhibits little effect. Indeed, sorbitol appeared to have a significantly lower delay than maltitol and sucrose on starch gelatinization. The impacts of the formulation were amplified at limiting water content as the different ingredients, especially starch and sweeteners, enter a competition to capture the little water available. The study of the priorities of the different ingredients on the water by the mixing cells aims to determine the origin of these differences observed in the case of sorbitol.

Ingredients’ Priorities on Water

The results obtained by calorimetry and X-ray diffraction on the doughs and model systems showed a different behavior of sorbitol compared to maltitol and to sucrose. The source of this difference is sought with the use of mixing cells in calorimetry. In this part, the degree of competition taking place between the starch in the flour and each of the sweeteners was investigated, with regard to the hygroscopicity of the sweeteners. To do this, the following four model systems were studied: F_50_W_50_; Su_50_W_50_; Ma_50_W_50_; So_50_W_50_; F_25_Su_25_W_50_; F_25_Ma_25_W_50_; F_25_So_25_W_50_. The study was carried out in isothermal mode in microcalorimetry at 20 °C.

The powder ingredients of the model system (flour and sweeteners) were placed in the lower compartment of the mixing cell, while the water was introduced into the upper compartment using a syringe, the valve being closed. The separation valve of the two compartments was opened after calorimeter equilibrium was reached at 20 °C, and the system was homogenized using the cell agitator to ensure correct distribution of the water. Resulting thermograms are presented in Figure 10.

When adding water to the flour alone, an exotherm immediately appeared. This exotherm is the manifestation of the energy required for sorption, also known as the heat of sorption [15].

In water-sweetener mixtures, when water came into contact with sugar or its substitutes (data not shown), an endotherm appeared. This endotherm corresponds to the dissolution of the solute (sugar, maltitol or sorbitol). The energy required for solubilization is taken from the environmental energy, the solution cools and this phenomenon is manifested by the presence of an endotherm.

In the same way, the mixtures with three components (flour-sweetener-water) were characterized. The wheat flour and the sweetener were mixed very carefully with a mortar to obtain a homogeneous system, and the water was then added in the mixing cell. The results in Figure 10 show different behavior depending on the composition of the model systems. 

The thermograms for the two systems F_25_Su_25_W_50_ and F_25_Ma_25_W_50_ first presented an exotherm, followed by an endotherm. This first exotherm corresponds to the rearrangement of amylopectin molecules in the starch grain in the presence of sorbed water, while the endotherm is associated to the dissolution of sugar and maltitol, as mentioned above. Conversely, sorbitol exhibited a completely atypical behavior. Only the endotherm from sorbitol dissolution was observed. No exotherm appeared. Thus, at limiting water content, starch does not seem to have a priority access to water in the presence of sorbitol, unlike maltitol and sucrose. 

Based on these observations, it is likely that the sorbitol is completely solubilized in the dough before the water enters the starch grain. The water-sorbitol solution behaves as a single plasticizer and penetrates into the grain. In the case of sucrose and maltitol, some of the water goes directly into the grain while the sweeteners dissolve in what remains of the water. The solution obtained is concentrated and more viscous; it therefore enters with more difficulty into the grain. Higher temperatures are then necessary to lower the solution viscosity and allow the plasticizer to penetrate into the starch grain. The starch then melts at a higher temperature, as observed in the model system and dough.

## 4. Conclusions

This work investigated the effects of the complete replacement of sucrose by two polyols on starch melting in biscuit dough. To this end, both doughs and model systems were studied at different water contents with wide-angle X-ray diffraction, calorimetry and environmental scanning microscopy. The study of model systems showed that the starch gelatinization temperature increased with decreasing water content and in the presence of each of the sweeteners. However, the extent of this shift was less marked in the case of sorbitol. At limiting water content, the enthalpy of starch melting was higher for sorbitol compared to sucrose and maltitol. These results were also found in the dough. The dough study showed that, indeed, at real water content, the substitution of sucrose by sorbitol induced a shift of the starch melting towards lower temperatures. The study of the different ingredients’ priorities on water in model media using mixing batch cells revealed a different behavior of sorbitol. Sorbitol exhibited priority over water compared to flour, while a competition occurred in the case of sucrose and maltitol. Therefore, in the dough, the sorbitol could solubilize quickly and enter with the water into the starch granule, allowing a higher degree of gelatinization before water evaporation during baking. In the case of maltitol and sucrose, part of the water might penetrate the starch granule, while the other part could form a syrup with the solubilized sweeteners. The higher viscosity of the syrup would make it difficult to penetrate the starch grain and delay this phenomenon towards higher temperatures. With less water available, gelatinization at the end of baking would then remain very partial. Depending on the water content of the product, this phenomenon may become substantial and affect certain sensory properties in the biscuit, such as its hardness, but also nutritional characteristics (glycemic index). It therefore seems relevant to consider this effect in a reformulation strategy. It is also interesting to note that the impacts of the formulation on the dough-biscuit transition and on the behavior of the starch during baking also materialized in the finished product. Indeed, the doughs studied in this investigation were baked in order to characterize the macroscopic and sensory characteristics of the biscuits obtained in another study [24]. Thus the sugar-free biscuits proved to be less prone to spontaneous checking and cracking phenomena, in particular the sorbitol biscuits whose spatial distribution of the water proved to be more homogeneous than for other formulations. On the other hand, the sensory analysis revealed that the substituted biscuits were less appreciated than the biscuit with sucrose, the biscuit with sorbitol being the least well-rated, while the biscuit with maltitol proved to be a satisfactory alternative.

## Figures and Tables

**Figure 1 foods-12-00607-f001:**
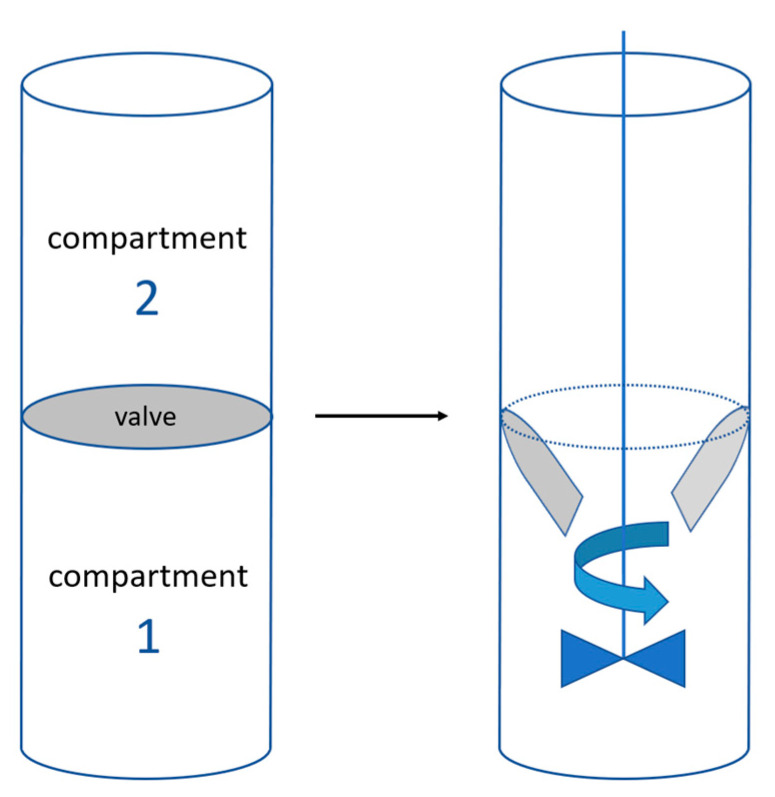
“Mixing batch” cell mechanism, including two compartments separated with a valve. Flour and sweeteners were placed in compartment n°1, and water in compartment n°2. When the equilibrium state is reached in the DSC, the valve is opened, the liquid enters into contact with the powders and the mixture is homogenized.

**Figure 2 foods-12-00607-f002:**
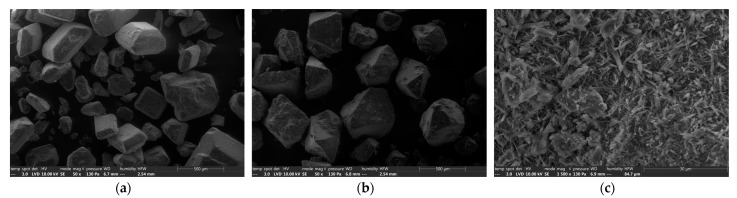
Scanning electron microscope (SEM) analyses of different sweeteners. (**a**): sucrose crystals with geometric shape ×50, (**b**): maltitol crystals with geometric shape ×50, (**c**): aggregated fine sorbitol flakes ×500.

**Figure 3 foods-12-00607-f003:**
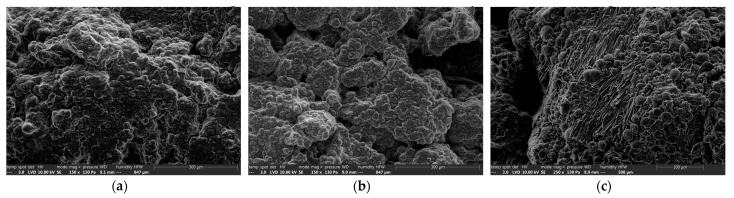
Environmental Scanning electron microscope (ESEM) analyses of the different loose doughs. (**a**): control dough exhibiting smooth starch granules and sucrose crystals embedded in a compact matrix, ×150, (**b**): maltitol dough exhibiting smooth starch granules and maltitol crystals embedded in a powdery matrix, ×150, (**c**): sorbitol dough magnified at ×250 to reveal localized sorbitol flakes among smooth starch granules embedded in a slightly powdery matrix.

**Figure 4 foods-12-00607-f004:**
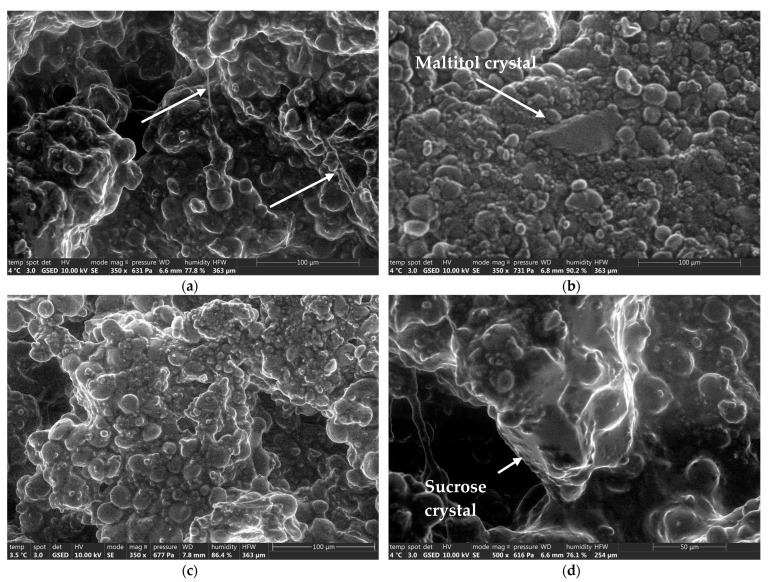
Environmental Scanning electron microscope (ESEM) analyses of dough pieces resulting from the different formulations. (**a**): sucrose dough piece exhibiting an aerated matrix and gluten filaments (white arrows) G × 350, (**b**): maltitol dough exhibiting a compact matrix with embedded maltitol crystals (white arrow) G × 350, (**c**): sorbitol dough exhibiting a compact matrix, no sorbitol flakes were observed at this stage of the process G × 350, (**d**): ESEM zoom on sucrose crystal embedded in the dough matrix (white arrow) G × 500.

**Figure 5 foods-12-00607-f005:**
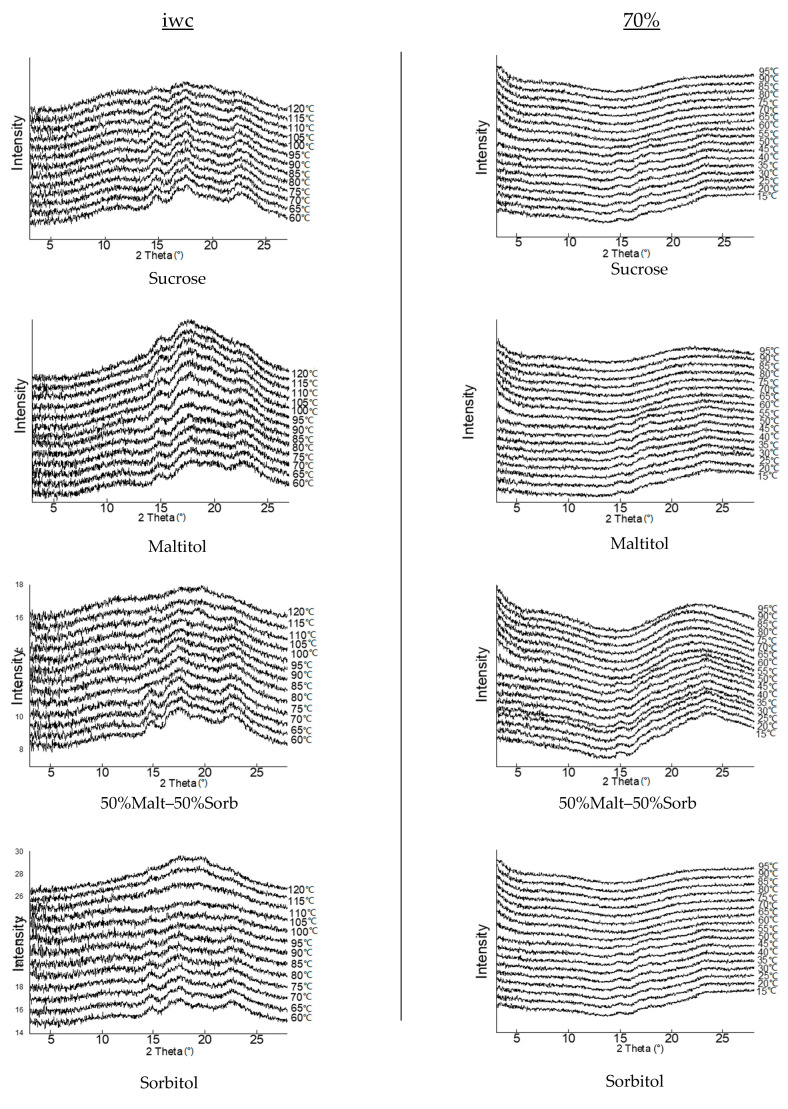
WAXS diffractograms of control and reformulated doughs at initial water content (iwc) and in excess water (70%) during baking kinetics from +60 °C to +120 °C. Diffraction patterns are offset for clarity.

**Figure 6 foods-12-00607-f006:**
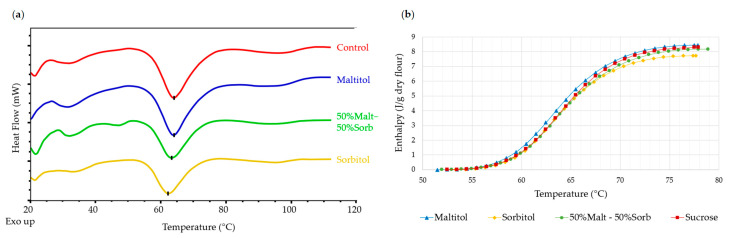
Examples of thermograms obtained by calorimetry with the substituted and reference doughs (**a**) and mean enthalpy of starch gelatinization as a function of temperature in control and no-added-sugar biscuit doughs (**b**) (*n* = 3). Doughs in water excess (70%).

**Figure 7 foods-12-00607-f007:**
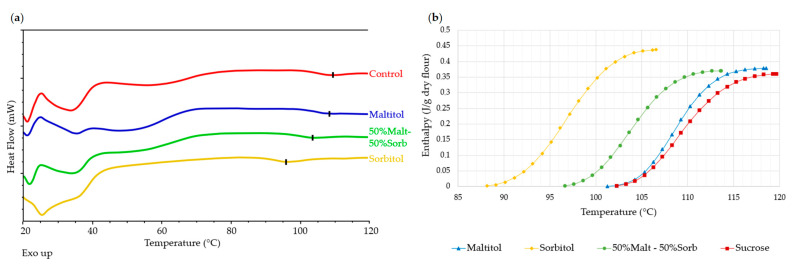
Examples of thermograms obtained by calorimetry with the substituted and reference doughs (**a**) and mean enthalpy of starch melting as a function of the temperature in the control and no-added-sugar biscuit doughs (**b**) (*n* = 3). Dough at initial water content.

**Figure 8 foods-12-00607-f008:**
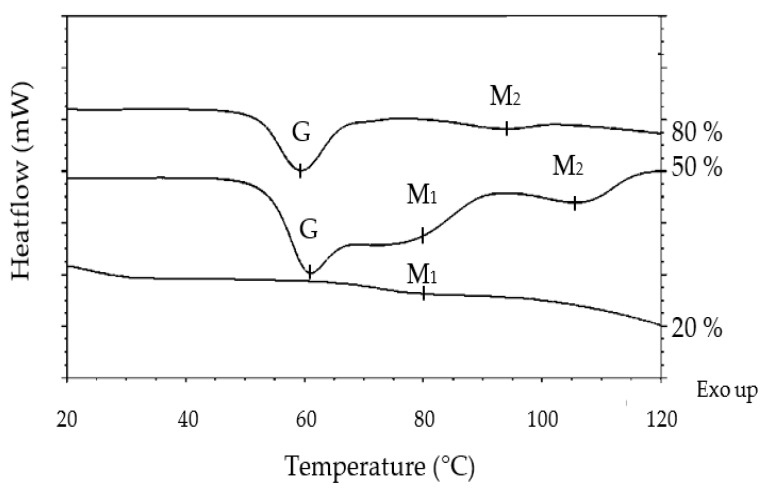
Representative DSC thermograms of flour-water mixtures at different water contents. Thermograms are offset for clarity.

**Figure 9 foods-12-00607-f009:**
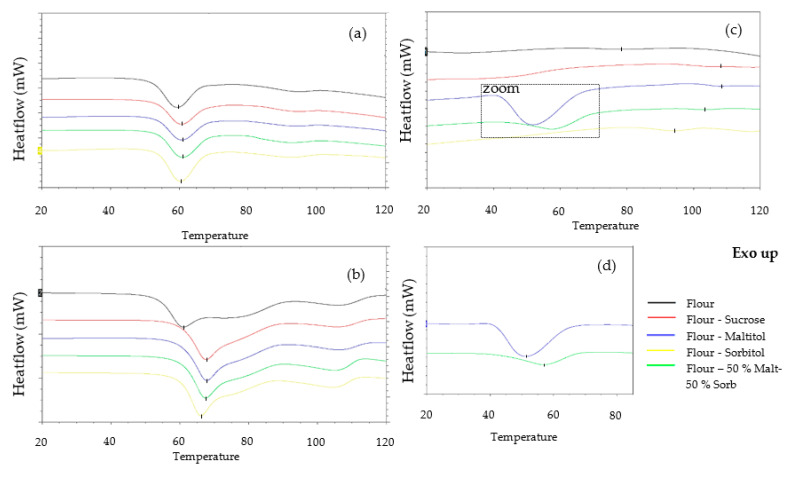
Representative DSC thermograms of model systems at different water contents. (**a**): excess water—G endotherms (80% d.b.); (**b**): intermediate water content—G+M_1_ endotherms (50% d.b.); (**c**): limited water content—M_1_ endotherms (20% d.b.); (**d**)**:** focus on endotherms observed around 50 °C in formulations containing maltitol. Thermograms are offset for clarity. The flour-sweetener ratio used in all model systems is the same as in biscuit dough, i.e., 3.35 in dry basis.

**Figure 10 foods-12-00607-f010:**
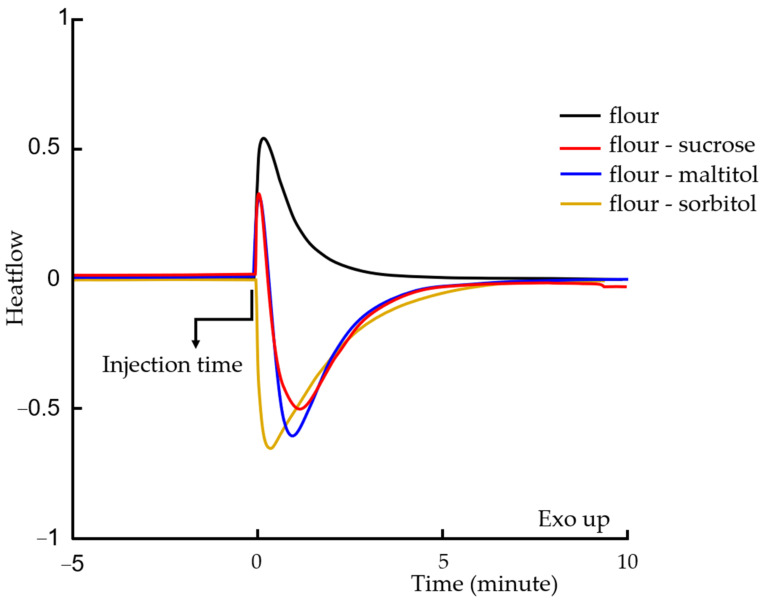
Representative thermograms of flour-water, flour-sucrose-water, flour-maltitol-water and flour-sorbitol-water mixtures at 20 °C (exotherm up).

**Table 1 foods-12-00607-t001:** Biscuit dough formulations (%, wet basis (w.b.)).

Ingredients (%, w.b.)	Control	Sorbitol	Maltitol	50% Maltitol–50% Sorbitol
Wheat flour	63.4	63.4	63.4	63.4
Sucrose	16.3	-	-	-
Sorbitol	-	16.3	-	8.15
Maltitol	-	-	16.3	8.15
Butter	13.3	13.3	13.3	13.3
Water	6.1	6.1	6.1	6.1
Salt	0.5	0.5	0.5	0.5
Leavening agents (NaHCO_3_ + Na_2_H_2_P_2_O_7_)	0.4	0.4	0.4	0.4

**Table 2 foods-12-00607-t002:** Water contents of ingredients (%, w.b.).

Ingredients	Water Content (% w.b.)
Wheat flour	14.66 ± 1.05
Sucrose	0.05 ± 0.01
Sorbitol	0.04 ± 0.00
Maltitol	0.22 ± 0.02
Butter	16.72 ± 0.09

**Table 3 foods-12-00607-t003:** Model system formulations with flour, water and sweeteners.

Model System (w.b.)	Ingredients	System Water Content (%)
1-F_2.32_ W_7.68_	Flour Water	80%
2-F_5.79_ W_4.21_	Flour Water	50%
3-F_9.26_ W_0.74_	Flour Water	20%
4-F_1.78_(Su_0.46_ W_7.76_)	Flour Sucrose Water	80%
5-F_4.46_(Su_1.15_ W_4.39_)	Flour Sucrose Water	50%
6-F_7.13_(Su_1.84_ W_1.03_)	Flour Sucrose Water	20%
7- F_1.78_(Ma_0.46_ W_7.76_)	Flour Maltitol Water	80%
8-F_4.46_(Ma_1.15_ W_4.39_)	Flour Maltitol Water	50%
9-F_7.13_(Ma_1.84_ W_1.03_)	Flour Maltitol Water	20%
10- F_1.78_(So_0.46_ W_7.76_)	Flour Sorbitol Water	80%
11- F_4.46_(So_1.15_ W_4.39_)	Flour Sorbitol Water	50%
12- F_7.13_(So_1.84_ W_1.03_)	Flour Sorbitol Water	20%
13- F_1.78_(Ma_0.23_So_0.23_ W_7.76_)	Flour Maltitol Sorbitol Water	80%
14- F_4.47_(Ma_0.57_So_0.57_ W_4.39_)	Flour Maltitol Sorbitol Water	50%
15- F_7.13_(Ma_0.92_So_0.92_ W_1.03_)	Flour Maltitol Sorbitol Water	20%

**Table 4 foods-12-00607-t004:** Water contents of doughs (%, w.b.).

Ingredients	Water Content (% w.b.)
Control	17.0 ± 0.1 ^a^
Maltitol	17.1 ± 0.2 ^a^
50–50	17.1 ± 0.6 ^a^
Sorbitol	17.3 ± 0.6 ^a^

a—Difference between each group is not statistically significant.

**Table 5 foods-12-00607-t005:** Starch gelatinization characteristics in doughs at initial water content (iwc) and in water excess (70%). Different letters within the same column for the same water content reflect significant differences (Duncan test, α = 5%).

		Enthalpy (J/g Dry Flour)	T_onset_ (°C)	T_peak_ (°C)	T_end_ (°C)
Control	iwc	0.30 ± 0.03 ns	103.0 ± 0.3 a	108.6 ± 0.3 a	118.9 ± 0.6 a
Maltitol	iwc	0.33 ± 0.01 ns	102.9 ± 0.2 a	108.4 ± 0.2 a	117.4 ± 0.5 a
50%Malt–50%Sorb	iwc	0.29 ± 0.03 ns	97.5 ± 0.2 b	103.9 ± 0.3 b	112.9 ± 1.4 b
Sorbitol	iwc	0.38 ± 0.04 ns	89.6 ± 0.2 c	96.6 ± 0.4 c	105.5 ± 0.8 c
Control	70%	8.2 ± 0.4 ns	56.7 ± 0.3 ns	63.8 ± 0.2 ns	78.2 ± 0.8 ns
Maltitol	70%	8.4 ± 0.3 ns	56.4 ± 0.5 ns	63.5 ± 0.4 ns	78.5 ± 0.9 ns
50%Malt–50%Sorb	70%	7.8 ± 0.3 ns	56.7 ± 0.1 ns	63.5 ± 0.1 ns	77.4 ± 1.1 ns
Sorbitol	70%	7.6 ± 0.3 ns	56.4 ± 0.3 ns	62.7 ± 0.3 ns	77.1 ± 0.7 ns

ns: difference is statistically not significant.

**Table 6 foods-12-00607-t006:** Starch gelatinization enthalpy (J/g dry flour) and temperatures (°C) (*n* = 3). Different letters within the same column for the same water content reflect significant differences (Duncan test, α = 5%).

Model System	Enthalpy (J/g _dry flour_)	T_onset_ (°C)	T_peak_ (°C)	T_end_ (°C)
1—F_2.32_W_7.68_	6.0 ± 0.4 ns	53.3 ± 0.8 ns	60.3 ± 1.4 ns	71.0 ± 1.7 ns
4—F_1.78_(Su_0.46_ W_7.76_)	6.9 ± 0.1 ns	54.4 ± 0.8 ns	61.5 ± 1.3 ns	74.0 ± 1.1 ns
7—F_1.78_(Ma_0.46_ W_7.76_)	6.4 ± 0.9 ns	54.5 ± 0.9 ns	61.5 ± 1.4 ns	71.8 ± 1.6 ns
10—F_1.78_(So_0.46_ W_7.76_)	6.7 ± 0.5 ns	54.0 ± 0.1 ns	60.3 ± 0.1 ns	69.6 ± 0.2 ns
13—F_1.78_(Ma_0.23_So_0.23_ W_7.76_)	7.9 ± 0.1 ns	54.22 ± 0.01 ns	60.9 ± 0.1 ns	72.0 ± 0.9 ns
2—F_5.79_W_4.21_	7.5 ± 0.3 b	54.3 ± 0.2 b	61.1 ± 0.3 d	90.2 ± 0.4 a
5—F_4.46_(Su_1.15_ W_4.39_)	8.6 ± 0.2 a	61.0 ± 0.4 a	67.9 ± 0.5 ab	88.3 ± 0.5 b
8—F_4.46_(Ma_1.15_ W_4.39_)	8.7 ± 0.3 a	61.1 ± 0.4 a	67.9 ± 0.3 a	88.4 ± 0.3 b
11—F_4.46_(So_1.15_ W_4.39_)	8.3 ± 0.2 a	60.3 ± 0.2 a	65.9 ± 0.2 c	85.9 ± 0.4 c
14—F_4.47_(Ma_0.57_So_0.57_ W_4.39_)	8.5 ± 0.1 a	60.7 ± 0.4 a	67.0 ± 0.4 b	87.4 ± 0.6 b
3—F_9.26_W_0.74_	0.21 ± 0.02 ns	70.8 ± 1.2 c	79.9 ± 2.2 c	95.0 ± 2 c
6—F_7.13_(Su_1.84_ W_1.03_)	0.11 ± 0.02 ns	99.9 ± 1.9 a	105.1 ± 0.8 a	113.0 ± 1.4 a
9—F_7.13_(Ma_1.84_ W_1.03_)	0.4 ± 0.3 ns	99.6 ± 3.3 a	105.9 ± 1.8 a	112.7 ± 1.9 a
12—F_7.13_(So_1.84_ W_1.03_)	1.2 ± 0.7 ns	84.1 ± 2.2 b	92.7 ± 2.1 b	102.2 ± 0.5 b
15—F_7.13_(Ma_0.92_So_0.92_ W_1.03_)	0.5 ± 0.2 ns	96.7 ± 0.6 a	102.7 ± 0.6 a	110.1 ± 0.8 a

ns: difference is not statistically significant.

## Data Availability

Data are available on request from the corresponding author except for sensory analysis data unavailable due to privacy constraints.

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
