# Peer review of "Effects of Sucrose Replacement by Polyols on the Dough-Biscuit Transition: Understanding by Model Systems"

_foods, 2023, doi:10.3390/foods12030607_

Round 1

Reviewer 1 Report

The article entitled Effects of sucrose replacement by polyols on the dough-biscuit transition: understanding by model systems has an interesting subject but the authors must improve the presentation of their findings.

In general, the authors have not presented deeply the scientific observation. They improve their comments based on other findings from literature.

I would recommend to add a PCA to article too see if the analysis made on the subject creates a clusterisation of the samples.

Author Response

Dear reviewer n°1,

Thank you very much for your comments and suggestions as well as for the time you have devoted to our work. Please see the attachment with the answers and modifications made as a result of your feedback.

Kind regards,

M. Roze

Reviewer 2 Report

The manuscript describes the study about the impacts of the complete substitution of sucrose by maltitol and/or sorbitol on the dough-crumb transition in biscuits. Starch gelatinization/melting were studied at different moisture contents in the biscuit dough and in the model systems by X-ray diffraction, differential scanning calorimetry and by environmental scanning electron microscopy.

The study may be interesting for readers. The article is well organized and contains all the necessary elements. The conclusions are consistent with the evidence and arguments presented.

Although the concept of the study is appropriate, I would recommend before the acceptance of the manuscript the baking tests with these types of dough to study differences in physical and sensory properties.

Author Response

Dear reviewer n°2,

Thank you very much for your comments and your very relevant suggestion to link our findings with resulting biscuit quality, as well as for the time you have devoted to our work. Please see the attachment with the answers and modifications made as a result of your feedback.

Kind regards,

M. Roze

Author Response

Dear reviewer n°3,

Thank you very much for your comments and suggestions as well as for the time you have devoted to our work. Please see the attachment with the answers and modifications made as a result of your feedback.

Kind regards,

M. Roze

Round 2

Reviewer 1 Report

Accept as it is

Author Response

Dear reviewer,

thank you for your comment.

Kind regards,

Mathilde Roze